# *Schisandra henryi*—A Rare Species with High Medicinal Potential

**DOI:** 10.3390/molecules28114333

**Published:** 2023-05-25

**Authors:** Karolina Jafernik, Halina Ekiert, Agnieszka Szopa

**Affiliations:** Chair and Department of Pharmaceutical Botany, Faculty of Pharmacy, Collegium Medicum, Jagiellonian University, Medyczna 9 Street, 30-688 Kraków, Poland; karolina.jafernik@doctoral.uj.edu.pl (K.J.); mfekiert@cyf-kr.edu.pl (H.E.)

**Keywords:** *Schisandra henryi*, *Schisandra chinensis*, chemical composition, dibenzocyclooctadiene lignans, Schisandra lignans, biotechnological research, traditional Chinese medicine, therapeutical potential

## Abstract

*Schisandra henryi* (Schisandraceae) is a plant species endemic to Yunnan Province in China and is little known in Europe and America. To date, few studies, mainly performed by Chinese researchers, have been conducted on *S. henryi*. The chemical composition of this plant is dominated by lignans (dibenzocyclooctadiene, aryltetralin, dibenzylbutane), polyphenols (phenolic acids, flavonoids), triterpenoids, and nortriterpenoids. The research on the chemical profile of *S. henryi* showed a similar chemical composition to *S. chinensis*—a globally known pharmacopoeial species with valuable medicinal properties whichis the best-known species of the genus Schisandra. The whole genus is characterized by the presence of the aforementioned specific dibenzocyclooctadiene lignans, known as “Schisandra lignans”. This paper was intended to provide a comprehensive review of the scientific literature published on the research conducted on *S. henryi*, with particular emphasis on the chemical composition and biological properties. Recently, a phytochemical, biological, and biotechnological study conducted by our team highlighted the great potential of *S. henryi* in in vitro cultures. The biotechnological research revealed the possibilities of the use of biomass from *S. henryi* as an alternative to raw material that cannot be easily obtained from natural sites. Moreover, the characterization of dibenzocyclooctadiene lignans specific to the Schisandraceae family was provided. Except for several scientific studies which have confirmed the most valuable pharmacological properties of these lignans, hepatoprotective and hepatoregenerative, this article also reviews studies that have confirmed the anti-inflammatory, neuroprotective, anticancer, antiviral, antioxidant, cardioprotective, and anti-osteoporotic effects and their application for treating intestinal dysfunction.

## 1. Introduction

The genus Schisandra (family Schisandraceae) currently includes 25 plant species [1,2], of which *Schisandra chinensis* Turcz. Baill. is the best-known species. *S. chinensis* is a pharmacopoeial medicinal plant that occurs naturally in Southeast Asia and is now cultivated in many temperate countries. The medicinal raw material of this plant is the fruit, i.e., *Schisandrae chinensis fructus*, the monographs of which are available in the pharmacopoeias of Asian countries such as China [3], Japan [4], and Korea [5]; moreover, in the European Pharmacopoeia [6]; United States Pharmacopoeia [7]; Russian Pharmacopoeia [8]; and in the International Pharmacopoeia published by WHO [9].

The therapeutic effects of *S. chinensis* fruit extracts reported thus far mainly include hepatoprotective, hepatoregenerative, anticancer, adaptogenic, immunostimulating, and anti-inflammatory activities [10,11].

*Schisandra sphenanthera* Rehder & Wilson is another species of the genus Schisandra that is well known in some areas of East Asia. Because of morphological similarity, *S. sphenanthera* is frequently confused with *S. chinensis*. Like *S. chinensis*, the monographs of *S. sphenanthera* are available in the Chinese Pharmacopoeia [3,12]. The medicinal raw material of this plant is also its fruit, i.e., *Schisandrae sphenanthera fructus*. *S. sphenanthera* fruit extracts are widely used in China; the lignan isolated from these extracts, namely schisanterin A, is the active ingredient of the standardized drug “Wuzhi tablet”, which has a hepatoprotective effect [13,14,15].

This high healing potential of *S. chinensis* and *S. sphenanthera* is because of their unique chemical composition. A characteristic of the Schisandraceae family is the presence of dibenzocyclooctadiene lignans (“Schisandra lignans”) as the main, specific group of compounds in plant extracts. These compounds are currently being intensively researched by scientists from Asian countries and several European research institutions [16,17,18].

Other species of the genus Schisandra are practically unknown in European countries. Most often, they are endemics that occur in a given province and have only been used in their areas. Most often there are small mentions in the TMC of how they were used by the local community for medicinal purposes [12].

The first little-known species is *Schisandra rubriflora* Rehd. and Will. Naturally, *S. rubriflora* occurs in Sichuan Province in China. It is described in TCM as a tonic and sedative. To this day, the fruit of *S. rubriflora* is considered a local delicacy. The analysis of the phytochemical profile showed the presence of dibenzocyclooctadiene lignans, which are characteristic only for this species: rubrischisantherin and rubrilignans A and B [12,19,20].

Another unknown species in Europe is *Schisandra grandiflora* Hook. F. & Thoms. It naturally occurs in the area of India (Qinling Mountains), where the fruit is used by the local population as a hepatoprotective agent and a delicacy due to its clove flavor. A small number of studies confirmed the presence of compounds from the dibenzocyclooctadiene, tetrahydrofuran, dibenzylbutane, diaryldimethylbutane, and tetralin group of lignans and compounds from the group of triterpenoids [21,22,23,24].

*Schisandra propinqua* (Wall.) Baill. is another unknown species in European countries, but quite widespread in southwestern China. There is quite a lot of information in the literature about the use of *S. propinqua* in TCM. The most commonly used extracts from rhizomes and stems have been administered orally or topically. *S. propinqua* has been used as an analgesic, also for stomach and liver problems. The analysis of the chemical composition of *S. propinuqa* primarily shows the presence of dibenzocyclooctadiene lignans, as well as compounds from the group of terpenoids and their derivatives [25,26,27,28].

A review has already been written by us, on the comparative analysis of three Schisandra species (*S. rubriflora, S. sphenanthera, S. henryi*), but this review focuses primarily on the *S. henryi* species [19].

*Schisandra henryi* C.B. Clarke is a species related to *S. chinensis* and *S. sphenanthera*; however, it is not well known. The healing properties of *S. henryi* are known in Far East countries, and this plant species has been used in traditional Chinese medicine (TCM) [29,30]. However, this plant species is little known or completely unknown in European countries and other parts of the world. *S. henryi* is a dioecious climber vine with characteristic yellow-orange flowers [1,11,31]. The available scientific studies show that the chemical composition of *S. henryi* is similar to that of other *Schisandra* species. The chemical constituents of *S. henryi* are mainly dominated by dibenzocyclooctadiene lignans, followed by aryltetralin and dibenzylbutane lignans. The high contents of triterpenoids and polyphenolic compounds have also been confirmed in *S. henryi* [32].

The present study aimed to review the available scientific literature on *S. henryi*, its botanical and ecological characteristics, the results of phytochemical analyses, and the biological properties of individual compounds detected in *S. henryi*, particularly dibenzocyclooctadiene lignans.

## 2. Methodology

The literature contained in this article has been appropriately collected according to the exclusion and inclusion criteria. In the first part, criteria were defined, which consisted of excluding articles that were not documents and guidelines (e.g., book reviews or commentaries). It was assumed that the main area of interest would be the chemical composition and biological activity, in particular, of dibenzocyclooctadiene lignans isolated from *S. henryi*. The searchable databases used in this publication are PubMed/MEDLINE, Web of Science, SCOPUS, Wiley Online Library, Google Scholar, Taylor & Francis Online, and Science Direct/ELSEV-IER, EBSCO Discovery Service (EDS). Articles published between 1961 and 2022 were searched in these databases. The following words were used in the search: “*Schisandra henryi*”, “chemical composition of *Schisandra henryi*” “Biological activity of dibenzocyclooctadiene lignans” or “Activity of dibenzocyclooctadiene lignans” or “Schisandra lignans”, “*Schisandra henryi* lignans”, “*Schisandra henryi* terpenoids”, “*Schisandra* polyphenolic compounds”.

## 3. Morphology and Natural Habitats

*S. henryi* (Figure 1) is endemic plant to the Yunnan province in southwestern China [2,30]. Yunnan province is a mountainous area with many forests, rivers, and lakes. The area has a subtropical climate with warm summers and mild winters. In winter, the temperature does not fall below 10 °C. The area receives heavy rainfall, which is conducive to the development of specific vegetation. Optimal locations for the growth of *S. henryi* vines are shady thickets, forests, slopes, and places near mountain streams at an altitude of 500–1500 m above sea level. In the European climate, the plant is not fully resistant to temperatures below 0 °C; hence, it freezes in winter. In Poland, *S. henryi* is frequently grown as an ornamental plant. The twining shoots of *S. henryi* grow to a length of 3–6 m, with an annual growth of approximately 1–2 m. The shoots are initially light green and eventually become brown with visible, numerous lighter lenticels [2,33].

The flowering period of *S. henryi* is in May. The flowers of *S. henryi* are 1–2 cm in diameter, single, small, and yellow orange in color. Female flowers have more than 8 petals and 14–40 free stamens, while male flowers have 6–10 petals and 14–40 free stamens [33]. The fruits of *S. henryi* grow as red berries and form cluster-shaped infructescences [2]. A single fruit is 3 mm in length and approximately 3.5 mm in width and contains 1–2 kidney-shaped seeds. The leaves of *S. henryi* reach 7–15 cm in length and 4.5–7.5 cm in width [33]. The leaves are thin, glossy, elliptical ovoid in shape, and dark green in color. Young leaves are sharp, while the older ones are slightly serrated at the edges [1,2,33].

## 4. Chemical Composition

To date, few scientific studies have reported the phytochemical analysis of *S. henryi*. Most research studies have focused on the chemical composition of shoots and leaves. Lignans are the main group of compounds found in *S. henryi*. The extracts contain a particularly high content of dibenzocyclooctadiene lignans as well as aryltetralin and dibenzylbutane lignans [30,32,34].

Dibenzocyclooctadiene lignans are a very interesting group of secondary metabolites because of their specific chemical structure. They are derivatives of cis-*O*-hydroxycinnamic acid with a lactone structure. They are thought to be formed by metabolic transformations of shikimic acid. The pathways for the formation of individual groups of lignans have not been clearly explained to date. The biosynthesis pathway of dibenzocyclooctadiene lignans was presented by Umezewa et al. [35] (Figure 2). It is hypothesized that lignans may be formed by coupling propenylphenols due to their chemical structure—they do not have 9(9′)-oxygen attached. Their function in the plant is not yet fully understood. It is believed that these lignans have a protective effect and affect plant development. Because these lignans have a wide spectrum of activity, attempts have been made to extract them through chemical and biotechnological processes. However, because of the high cost and labor input, plant raw materials are still the most valuable source of obtaining dibenzocyclooctadiene lignans [35]. The first dibenzocyclooctadiene lignan to be isolated was schisandrin. The compound was isolated from *S. chinensis* seed oil by Kochetkow in 1961 [36]. As of today, the literature reports that about 150 lignans containing a dibenzocyclooctadiene skeleton have been isolated [37]. A large amount of research is focused on the creation of derivatives of dibenzocyclooctadiene lignans that would have similar biological activity. Bicyclol is a synthetic substance based on schisandrin C. In the initial stages of research on *Schisandra chinensis* extracts, chosen dibenzocyclooctadiene lignans were isolated and tested for hepatoprotective activity. Tests have shown that schisandrin C has promising activity in this manner. Despite many attempts, researchers have failed to elaborate on the full chemical synthesis procedure of schisandrin C, which has forced them to continue their research. It was found that appropriate changes in the positions of the methylene dioxide groups changing the length of the carboxylic acid to the biphenyl ring and changing the dicarnoxylate group to the hydroxyl group increase the effectiveness as well as bioavailability of the derivatives. In this way, bicyclol ((4,4′-dimethoxy-5,6,5′,6′-bis [methylenedioxy]-2-hydroxymethyl-2′-methoxycarbonyl biphenyl) was synthesized, which was registered as a drug by the Chinese Food and Drug Administration (FDA). The protocol of its synthesis as well as the production method on a large scale are covered by a patent [38,39,40,41]. Bicycol is approved by the Chinese FDA as a means of supporting the regenerative processes of the liver, especially in people with abnormal ALT parameters associated with liver diseases. In addition to clinical studies on the therapeutic effect of bicyclol on liver cells, researchers are focusing on studying the anticancer activity of this compound. For now, the tests are in the initial phase, but the results are promising [35,42,43,44]. The structures of dibenzocyclooctadiene lignans found in *S. henryi* are shown in Figure 3.

The presence of lignans was confirmed in the leaves and shoots of *S. henryi*; these lignans belonged to the following groups: (1) dibenzocyclooctadiene: gomisin G, schisanterin A, benzylgomisin Q, deoxyschisanadrin, and schisandrin; (2) aryltetralin: wulignan A1 and A2, epiwulignan A1, enshicin, epienshicin, and dimethylwulignan A1; (3) dibenzylbutane: henricin A and B and isoanwulignan (Figure 4) [29,45,46]. In tests conducted by a team from the Department of Pharmaceutical Botany, the Jagiellonian University Medical College, the following compounds were detected: dibenzocyclooctadiene lignans: schisandrin, gomisin G, schisantherin A and B, deoxyschisandrin, and schisandrin C (Figure 3); phenolic acids: gallic, chlorogenic, neochlorogenic, caftaric, and caffeic; and flavonoids: hyperoside, rutoside, trifolin, quercitrin, quercetin, and kaempferol (Figure 5) [32].

The extracts from *S. henryi* shoots were also found to contain ganschisandrin—a tetrahydrofuran lignan (Figure 4) [46]. Apart from lignans, the shoots were found to contain triterpenoid compounds: henrischinin A, B, and C and schisanlactone B; acids: isoschisandronic, kadsuric, anvuweizic, schisandronic, and nigrnoic acid as well as 3-ethyl-nigranic acid [47,48]; and nortriterpenoids: henridilactones A-D and schiprolactone A (Figure 5) [49].

The extract from the leaves and stems of *S. henryi* contained as many as 11 compounds from the group of schinortriteprenoids—henridilactones E-O [50].

The fruits of *S. henryi* showed the presence of compounds from the group of lignans (schisantherin B, schisanhenol, and schisanhenrin) and terpenoids (kadsuric and schisanhenric acids) (Figure 5) [51].

The literature data and procedures which have thus far been elaborated are very diverse in terms of the extraction methodology and identification of individual *S. henryi* compounds. The identification of compounds is most often based on previous works describing the isolation and chemical structure elucidation of a given compound [29,32,48,49,50,52]. Table 1 summarizes the applied extraction methodology and the apparatus used to identify groups or individual compounds from the published papers focused on *S. henryi* (Table 1).

## 5. Reports on the Biological Activities

Liu et al. isolated the following compounds from the seeds of *S. henryi*: 2 triterpenoids—kadsuric acid and schisanhenrin and 13 lignans—schisanterin A and B, schisanhenol, deoxyschisandrin, epiwulignan A1, wulignan A1 and A2, schisandrone, henricin, enshicin, epienshicin methyl ether, epischisandrone, and enshcine. Wulignan A1 and A2, epiwulignan A1, and epischisandrone were found to have inhibitory activity against P-388 lymphoma cell lines [45,53].

Chen et al. isolated four compounds from the dried shoots of *S. henryi*, namely gomisin G, schisantherin A, benzoylgomisin Q, and isowulignan, and they then tested the biological activity of these compounds on DNA strand cleavage and the cytotoxic activity in leukemia cell lines and HeLa cells (cervical cancer line) in vitro. Gomisin G in the presence of Cu^2+^ ions showed strong DNA cleavage activity at 50 μg/mL concentration, with more than 50% relaxation of supercoiled DNA. The other compounds showed no activity. In in vitro tests on cell lines, gomisin G exhibited the highest cytotoxic effect (IC_50_ = 5.51 μg/mL) on leukemia and HeLa cell lines. Schisantherin A and benzoylgomisin Q showed a moderate cytotoxic effect on leukemia cells (IC_50_ = 55.1 and 61.2 μg/mL, respectively). Benzoylgomisin Q showed a moderate cytotoxic effect (IC_50_ = 61.2 μg/mL) on HeLa cells, while schisantherin A did not affect these cells [52].

Jafernik et al., from the Department of Pharmaceutical Botany, the Jagiellonian University Medical College, studied the antioxidant and anti-inflammatory activities of *S. henryi* leaf extracts. Antioxidant tests were conducted using the CUPRAC, FRAP, and DPPH methods, while the anti-inflammatory activity was determined using the method for the inhibition of enzyme activity: 15-LOX, COX-1, COX-2, and sPLA2. The total polyphenol content was determined by the Folin–Ciocalteu spectrophotometric method. The total content of polyphenols in leaf extracts was 277 nmol/Gal/mg dry matter (DM). The antioxidant activity of leaf extracts according to the different methods was as follows: the CUPRAC method: 67 TE nmol/mg DM; the FRAP method: 24 TE nmol/mg DM; and the DPPH method: 53 TE nmol/mg DM. The anti-inflammatory effect (percent inhibition) of leaf extracts was as follows: sPLA2: 19%, 15-LOX: 26%, COX-1: 70%, and COX-2: 33% [32].

He et al. isolated 12 schinortriterpenoids (E-O henridilactones) from the stems and leaves of *S. henryi*, including 11 schinortriterpenoids, for the first time. The biological activity of the isolated compounds was then tested in terms of neuroprotective effects by inducing apoptosis with corticosterone in PC12 cells (a rat pheochromocytoma cell line used in neurological and toxicological studies). Four compounds, namely henridilactone E, H, N, and O, exhibited the strongest neuroprotective effect related to cell apoptosis inhibition. Additionally, henridilactone O increased the number of neurites [50].

## 6. Biological Activity of Chosen Dibenzocyclooctadiene Lignans

The review of scientific research showed that dibenzocyclooctadiene lignans are the most interesting group of compounds detected in *S. henryi* in terms of pharmacological activity. The best-studied compounds identified in *S. henryi* are schisandrin C, gomisin G, schisantherin A, and deoxyschisandrin. These compounds were confirmed to possess hepatoprotective, antioxidant, anti-inflammatory, and anticancer properties. These compounds also showed beneficial effects on nervous system functioning and could resolve issues associated with intestinal dysfunction (Table 2) [54,55,56,57,58,59,60,61].

### 6.1. Antioxidant Activity

Park et al. investigated the effect of schisandrin on MMP-1 (metalloproteinase-1) expression in UV-irradiated human HDF fibroblasts. Schisandrin inhibited lipid peroxidation in cells and decreased MMP-1 expression [54] (Table 2).

Lam et al. tested the use of schisandrin C and B as sunscreen compounds with antioxidant activity. Both lignans showed antioxidant activity, protected rat skin tissue from oxidative damage caused by UV radiation, and increased the concentration of reduced glutathione and α-tocopherol in cells. These lignans also increased the activity of antioxidant enzymes and malondialdehyde production [62] (Table 2).

Lu et al. tested the antioxidant activity of selected dibenoscyclooctadiene lignans. It has been shown that schisantherin B at a concentration (1 mM) effectively inhibits cysteine-induced iron-induced lipid peroxidation in rat liver microsomes. The study also showed a reduction in the production of superoxide anions in the xanthine/xanthine oxidase system. The comparative analysis showed that the effect of schisantherin B was much stronger than that of vitamin E at the same concentration (Table 2) [76].

### 6.2. Anti-Inflammatory Activity

Oh et al. showed that schisandrin C and gomisin N and J could suppress lipopolysaccharide (LPS)-induced inflammatory responses in RAW 264.7 mouse macrophage cell line (mouse macrophage cells). These compounds reduced nitric oxide (NO) production in stimulated RAW 264.7 cells, but did not show a cytotoxic effect. They also reduced the mRNA expression and secretion of proinflammatory cytokines. The mechanism of action for these compounds is based on blocking p38 mitogen-activated protein kinase (MAPK) [57] (Table 1).

Ci et al. confirmed that schisantherin A exhibits a strong anti-inflammatory effect by lowering the concentration of compounds involved in inflammatory processes, namely TNF-α (tumor necrosis factor α), IL-6 (interleukin 6), NO, and PGE2 (prostaglandin E2), induced by LPS. Schisantherin A was also found to reduce iNOS and COX-2 levels in RAW 264.7 macrophages. Signal transduction studies showed that schisanterin A significantly inhibited the expression of the extracellular signal-regulated kinase (ERK) phosphorylation protein, p38, and C-jun NH2-terminal kinase (JNK). Schisantherin A also inhibited the nuclear translocation of p65-NF-kB by IkB-α degradation. By using specific inhibitors of ERK, JNK, and p38, a previous study showed that schisantherin A can inhibit TNF-α mainly through the ERK pathway [66] (Table 2).

Liao et al. studied the anti-inflammatory effect of schisantherin A on interleukin-1β (IL-1β)-stimulated human chondrocytes with osteoarthritis. Human chondrocytes with osteoarthritis were pretreated with schisantherin A 1 h before administration. This study found that schisantherin A affects the production of NO, PGE2, iNOS, COX-2, and TNF-α. Schisantherin A also inhibited the IL-1β-induced production of NO, PGE2, and TNF-α in a dose-dependent manner. Moreover, the IL-1β-induced expression of MMP1, MMP3, and MMP13 was inhibited by schisantherin A. Furthermore, schisantherin A prevented the activation of NF-κB and MAPK by IL-1β [67] (Table 2).

Li et al. studied the anti-inflammatory effects of schisantherin A in neuroinflammatory conditions. Schisantherin A suppressed the inflammatory response in LPS-activated BV-2 microglia. It also inhibited NF-κB activation induced by LPS by affecting the IκB degradation and phosphorylation of IκB, IKK, PI3K/Akt, JNK, and p38 MAPK. Schisantherin A also showed indirect antioxidant activity by silencing the production of reactive oxygen species (ROS) and stimulating the expression of antioxidant enzymes (HO-1 and NQO-1) by stimulating Nrf2 activation pathways. Another study confirmed that schisantherin A exhibits an anti-neuroinflammatory effect by inducing Nrf2 through ERK phosphorylation [68] (Table 2).

Zhou et al. tested the anti-inflammatory effects of schisantherin A in acute respiratory distress syndrome, which involves the adhesion, activation, and sequestration of polymorphonuclear neutrophils and inflammatory damage to the alveolar capillary tissue membrane. This study was conducted in a mouse model of acute respiratory distress syndrome induced by LPS. The severity of lung injury was assessed 7 h after LPS administration. The results showed that the wet weight to dry weight ratio; myeloperoxidase activity; and the total number of cells, neutrophils, and macrophages in the bronchoalveolar lavage fluid were significantly reduced after schisanterin A treatment. Pretreatment with schisantherin A significantly alleviated LPS-induced histopathology and decreased levels TNF-α, IL-6, and IL-1β in bronchoalveolar fluid. Schisanterin A also suppressed LPS-induced phosphorylation of NF-κB p65, inhibitor kappa B-alpha (IκB-α), JNK, ERK, and p38. Previous studies have also confirmed the strong anti-inflammatory effects of schisantherin A, which resulted from blocking the activation of NF-kB and MAPK signaling pathways [69] (Table 2).

### 6.3. Anticancer Activity

Casarin et al. investigated the anticancer effect of deoxychisandrin and gomisin N on two human cancer cell lines: colon adenocarcinoma (LoVo) and ovarian adenocarcinoma (OV-2008). Lignans inhibited cell growth in a dose-dependent manner in both cell lines, but they induced different types of cell death. Deoxyschisandrin induced apoptosis in LoVo cells but not in OV-2008 cells, while gomisin N induced apoptosis in both cell lines. Both compounds caused cell growth arrest in the G2/M phase, which was correlated with tubulin polymerization [58] (Table 2).

Maharjan et al. confirmed the anticancer effect of gomisin G. They showed that gomisin G suppresses the viability of breast cancer cell lines: TNBC, MDA-MB-231, and MDA-MB-468. Gomisin G did not induce apoptosis but drastically inhibited AKT phosphorylation and downregulated retinoblastoma tumor suppressor protein (Rb) and phosphorylated Rb. It also decreased cyclin D1 level in a proteasome-dependent manner, thereby leading to cell cycle arrest in the G1 phase [77] (Table 1).

The same team extended their research on the anticancer effect of gomisin G on colorectal cancer cells. Gomisin G significantly inhibited the viability and production of LoVo cells. Gomisin G downregulated AKT phosphorylation, thereby suppressing the PI3K-AKT signaling pathway; it also induced apoptosis as demonstrated by annexin V staining and increased levels of cleaved poly-ADP ribose polymerase (PARP) and caspase-3 proteins. Gomisin G significantly accumulated cells in the sub-G1 phase, which represents apoptotic cells [78] (Table 2).

Wang et al. showed that schisantherin A has antiproliferative effects on gastric cancer cell lines MKN45 and SGC-7901. Schisantherin A induced cell cycle arrest at the G2/M phase and cell apoptosis and inhibited cell migration in MKN45 and SGC7901 gastric cancer cells. Schisantherin A induced ROS-dependent JNK phosphorylation with higher ROS production. The ROS scavenger JNKi NAC inhibitor caused schisantherin A-induced cell apoptosis and cell cycle arrest [59] (Table 2).

Chen et al. studied the biological activity of three lignans: gomisin G, schisantherin A, and benzoylgomisin Q. The authors demonstrated a significant cytotoxic effect of gomisin G on leukemia and HeLa cells (cervical cancer cells) in vitro with an IC50 (half of the maximum inhibitory concentration) value of 5.51 µg/mL against both cell lines. Schisantherin A and benzoylgomisin Q showed moderate cytotoxic activity against leukemia cells, with IC50 values of 55.1 and 61.2 µg/mL, respectively. Benzoylgomisin Q also showed cytotoxicity against HeLa cells, with an IC50 value of 61.2 µg/mL [70] (Table 2).

### 6.4. Antiviral Activity

Chen et al. conducted a study on the anti-HIV-1 effect of gomisin G. The EC50 (half of the maximum effective concentration) value was 0.006 µg/mL, and the therapeutic index (TI) was 300. Gomisin G inhibited HIV-1 replication, and according to the researchers, this inhibitory effect was due to the chemical structure of gomisin G with the appropriate position of the phenolic substituents on the hydroxyl groups [50] (Table 1).

Xu et al. tested six lignans, of which two, namely deoxyschisandrin and schisandrin B, showed antiviral activity. Both lignans exhibited a significant inhibitory effect on HIV-1 reverse transcriptase and viral replication. Their mechanism of antiviral activity is based on the selective inhibition of DNA polymerase associated with HIV-1 reverse transcriptase [63] (Table 2).

### 6.5. Neuroprotective Activity

Li et al. studied the effects of schisantherin A on cognition and neurodegeneration in mice with Alzheimer’s disease (AD) induced by Aβ1-42 (amyloid β-peptide). The authors found that the intracerebroventricular (ICV) administration of schisantherin A (at the doses of 0.01 and 0.1 mg/kg body weight) for 5 days significantly attenuated Aβ1-42-induced learning and memory impairment as assessed by the Y-maze test, the Morris Water Maze test, and the Shuttle Box test. Schisantherin A at the dose of 0.1 mg/kg restored some degree of superoxide dismutase (SOD) and glutathione peroxidase (GSH-Px) enzyme activity as well as the levels of Aβ1-42, malondialdehyde (MDA), and glutathione (GSH) in the hippocampus and cerebral cortex. Schisantherin A also improved histopathological changes in the hippocampus. These results suggest that schisantherin A may protect against cognitive deficits, oxidative stress, and Aβ1-42-induced neurodegeneration and serve as a potential agent for treating AD [71] (Table 2).

Sa et al. assessed the neuroprotective effects of schisantherin A in preventing Parkinson’s disease (PD). This study used SH-SY5Y cells (neuroblastoma cells) incubated with 1-methyl-4-phenylpyridinium ion (MPP(+)) and mice treated with 1-methyl-4-phenyl-1,2,3,6-tetrahydropyridine (MPTP). Schisantherin A treatment significantly inhibited MPP(+)-induced cytotoxicity in SH-SY5Y cells and provided significant protection against MPTP-induced loss of TH-positive dopaminergic neurons in a mouse model of PD. Western blotting assay showed that schisantherin A exerts a neuroprotective effect against MPP(+) by regulating two different pathways, including the CREB-mediated upregulation of Bcl-2 and activation of the PI3K/Akt survival signaling pathway [60] (Table 2).

Zhang et al. focused on the neuroprotective effect of schisantherin A. The protective effect of this compound against selective neuronal damage induced by the dopaminergic neurotoxin 6-hydroxydopamine (6-OHDA) was investigated in human SH-SY5Y cells and in a zebrafish model. Pretreatment with schisantherin A provided neuroprotection against 6-OHDA-induced cytotoxicity in SH-SY5Y cells and prevented the 6-OHDA-stimulated loss of dopaminergic neurons in zebrafish. Previous studies have shown that schisantherin A can regulate intracellular ROS accumulation and inhibit NO overproduction by reducing iNOS overexpression in SH-SY5Y cells treated with 6-OHDA. Schisantherin A was also confirmed to protect against the 6-OHDA-mediated activation of MAPK, PI3K/Akt, and GSK3β [72] (Table 2).

### 6.6. Hepatoprotective and Hepatoregenerative Activity

Zheng et al. investigated the effect of schisantherin A on ischemic and reperfusion-induced liver damage. The studies were conducted in male C57BL/6 mice in which sham laparotomy or liver reperfusion was induced after schisantherin A administration. The following parameters were assessed: liver function, histological damage, oxidative/nitrosative stress, inflammatory cell infiltration, cytokine production, cell apoptosis, cell autophagy, and activation/inhibition of intracellular signaling pathways with reperfusion. The treatment of mice with schisantherin A significantly preserved liver function, decreased histological damage, reduced oxidative/nitrosative stress, prevented inflammation, and inhibited cell apoptosis. The primary protective mechanism elicited by schisantherin A is presumed to be involved in inhibiting the MAPK signaling pathway [73] (Table 2).

Wang et al. also found beneficial effects of schisantherin A on the liver. Their study was conducted on a model of liver fibrosis in mice, which were gradually administered intraperitoneally thioacetoamide and schisantherin A (1, 2, and 4 mg/kg body weight) for 5 weeks. Schisantherin A significantly alleviated the pathological changes in the liver tissue induced by thioacetamide. It also reduced the levels of serum transaminases and hydroxyprolines and decreased the expression of α-smooth muscle actin (α-SMA) and collagen 1A1 (COL1A1) proteins in the liver tissue. Schisantherin A also reduced the levels of TNF-α, IL-1β, and IL-6 in serum and liver tissue and downregulated the expression of a target protein associated with the TAK1/MAPK and NF-κB pathways in the liver tissue. In vitro studies revealed that schisantherin A significantly inhibits TGF-β1-induced HCS-T6 cell proliferation and activation, downregulates TNF-α and IL-6 expression, and inhibits TGF-β1-induced TAK1 activation and subsequent expression of MAPK and proteins associated with the NF-κB signaling pathway [74] (Table 2).

### 6.7. Cardioprotective Activity

Chang et al. conducted studies that evaluated the effects of deoxyschisandrin and schisantherin A on the myocardium. The research was conducted on a rat model of myocardial ischemia–reperfusion injury. Male rats were administered deoxyschisandrin and schisantherin A (40 µmol/kg body weight) through the tail vein after 45 min of ischemia and 2 h of reperfusion. Cardiac function, infarct size, biochemical markers, histopathological changes, and apoptosis were assessed, and the mRNA expression level of gp91 phox in myocardial tissues was determined. Rat cardiomyocytes were initially treated with deoxyschisandrin and schisantherin A and then subjected to H2O2-induced damage. Both deoxyschisandrin and schisantherin A reduced arrhythmias, exerted a protective effect on heart function, significantly reduced myocardial infarction and MDA release, and increased SOD activity, which subsequently reduced myocardial damage. Furthermore, both compounds alleviated changes in myocardial histopathology and reduced cell apoptosis and caspase-3 activity in the myocardium; this exerted a protective effect on cardiomyocytes [64] (Table 2).

### 6.8. Supportive Activity in the Treatment of Intestinal Dysfunction

Xu et al. investigated the application of deoxyschisandrin for treating inflammatory bowel disease (IBD). Deoxychisandrin was administered to a mouse model of IBD and was found to affect the visceral sensitivity of the animals. The level of brain-derived neurotrophic factor (BDNF) was also determined in mice with intestinal hypersensitivity. It was observed that deoxyschisandrin inhibited the contraction of isolated smooth muscles, modulated the function of the gastrointestinal tract, and effectively reduced the disease activity index in the tested animals. The experiment also confirmed that deoxyschisandrin has an antidiarrheal effect [65] (Table 2).

In a colonic distension (CRD) experiment, visceral sensitivity was increased in the model group. However, the Abdominal Withdrawal Reflex (AWR) test showed that deoxyschisandrin reduced the AWR. This study revealed that schisandrin A significantly affects the reduction of visceral hypersensitivity in IBD mice. Immunohistochemical analysis and western blotting assay also demonstrated that BDNF protein expression was clearly increased in the colon of IBD mice. Following treatment with deoxyschisandrin, the BDNF protein expression in the colonic mucosa in IBD mice was decreased; this finding explains the mechanism of action for schisandrin A: a reduction in the intestinal sensitivity of mice by reducing BDNF expression in the colonic mucosa [65] (Table 2).

### 6.9. Anti-Osteoporotic Activity

Caichompoo et al. conducted an in vitro study to determine the effect of lignans on osteoblasts of the UMR 106 cell line. Deoxychisandrin, schisandrin, and γ-schisandrin increased cell proliferation and alkaline phosphatase activity in the osteoblasts, thus indicating their potential anti-osteoporotic activity [61] (Table 2).

## 7. Plant Biotechnology Research

In the Department of Pharmaceutical Botany of the Collegium Medicum of Jagiellonian University (UJCM), the in vitro cultures of *S. henryi* were initiated for the first time on a global scale (Figure 6). The culturing process was initiated using leaf buds from the male specimen of *S. henryi,* which were provided by the CLEMATIS company (Clematis Spółka z o.o., Pruszków, Poland). The cultures were grown on a Murashige and Skoog (MS) medium with the addition of the following plant growth regulators: 1 mg/L BA (6-benzyladenine) and 1 mg/L IBA (3-indolylbutyric acid). Various cultivation periods (10, 20, and 30 days) and concentrations of plant growth regulators (BA, IBA, and gibberellic acid (GA_3_)) were tested to optimize the conditions for the cultivation of stationary cultures—microshoot and callus (Figure 4). The biomass was found to contain dibenzocyclooctadiene lignans: schisandrin, gomisin G, schisantherin A and B, deoxyschisandrin, and schisandrin C; dibenzylbutane lignan: henricin B; aryltetralin lignans: wulignan A1 and A2, epiwulignan A1, enshicin, epienshicin, and dimethylwulignan A1; and triterpenoids: kadsuric acid and schisanhenric acid. Additionally, the content of selected dibenzocyclooctadiene lignans, phenolic acids, and flavonoids in the methanolic extracts of the biomass was estimated. The maximum contents of lignans, phenolic acids, and flavonoids were 873.71, 840.89, and 421.98 mg/100 g dry mass (DM), respectively. The highest content was noted for schisantherin B (maximum: 622.59 mg/100 g dry weight) and schisantherin A (maximum: 143.74 mg/100 g DM) among lignans; neochlorogenic acid (maximum: 472.82 mg/100 g DM) and caftaric acid (maximum: 370.81 mg/100 g DM) among phenolic acids; and trifolin (maximum: 138.56 mg/100 g DM) and quercitrin (maximum: 122.54 mg/100 g DM) among flavonoids. The content of lignans obtained in in vitro cultures was compared with the content determined in extracts from the leaves of the parent plant. Their content was found to be 13 times higher. Similarly, for phenolic acids—the content in in vitro cultures was more than six times higher, and one time higher for flavonoids [32]. Further studies on in vitro cultures of *S. henryi* are currently underway. Several experiments have been conducted to determine the biological activity profile of in vitro cultures. The production of secondary metabolites was also optimized and determined in microshoot cultures grown in special PlantForm^TM^ (PlantForm, Sweden) bioreactors (Figure 6), callus cultures, and suspension cultures have been stimulated by the addition of elicitors and biosynthetic precursors [unpublished].

## 8. Conclusions

*S. henryi* is an endemic species whose medicinal properties are recognized in TCM. The available scientific literature shows that the chemical composition of *S. henryi* is partly similar to that of *S. chinensis* because of the presence of specific dibenzocyclooctadiene lignans*. S. henryi* extracts also contain other compounds belonging to the group of terpenoids, polyphenols, and aryltetralin and dibenzylbutane lignans. Few studies on the biological activity of *S. henryi* leaf and shoot extracts have demonstrated their anticancer activity (against lymphoma, leukemia, and cervical cancer cell lines) as well as neuroprotective, antioxidant, and anti-inflammatory effects. The high therapeutic potential of this species is associated with the biological activity of compounds belonging to the group of dibenzocyclooctadiene lignans. Research in the field of plant biotechnology, which indicates a high biosynthetic potential of in vitro cultures of *S. henryi*, is also extremely interesting. The conducted research is based on the cultivation of *S. henryi* cultures on various types of solid, the addition of precursor compounds and elicitors, aimed at increasing the accumulation of secondary metabolites with a wide therapeutic potential.

The main problem related to the breeding of *S. henryi* is the acquisition of plant material for research, which, as previously mentioned, is related to the natural occurrence of *S. henryi*. Research on this species is most often carried out by Chinese research teams, due to the ease of obtaining material for research. Cooperation with the company Clematis, which deals with the cultivation of *S. henryi*, gives us the opportunity to conduct research on the species *S. henryi* in European countries.

In addition, the little knowledge in European countries about *S. henryi* and other species of the genus Schisandra is a big problem, which is associated with a small amount of research on the biological activity of *S. henryi*.

The research team from the Department of Pharmaceutical Botany of the Jagiellonian University Medical College focuses on continuing research on the biological activity of *S. henryi* extracts, both from the parent plant material and material obtained by biotechnology methods. Comparative analyzes are carried out to answer the question of whether the material obtained from in vitro cultures has a greater therapeutic potential than the material from the parent plant.

On the basis of this review, *S. henryi,* as one of the least-known representatives of the genus Schisandra, certainly deserves greater attention from scientists worldwide to better understand its chemical composition and pharmacological properties.

## Figures and Tables

**Figure 1 molecules-28-04333-f001:**
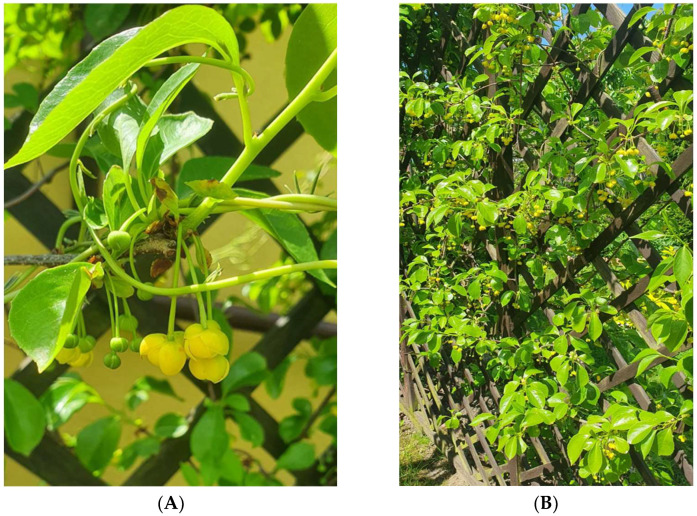
Morphological appearance of *S. henryi* (Garden of medicinal plants of the Faculty of Pharmacy, Collegium Medicum, Jagiellonian University, Kraków, Poland); (**A**)—flowers; (**B**)—plant habit.

**Figure 2 molecules-28-04333-f002:**
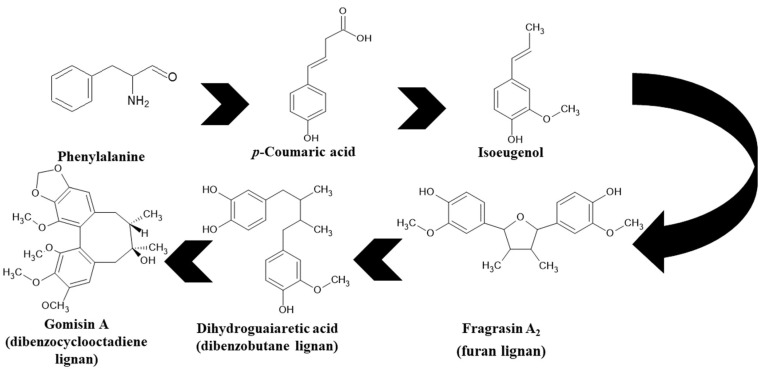
A suggested biosynthetic pathway of dibenzocyclooctadiene lignans.

**Figure 3 molecules-28-04333-f003:**
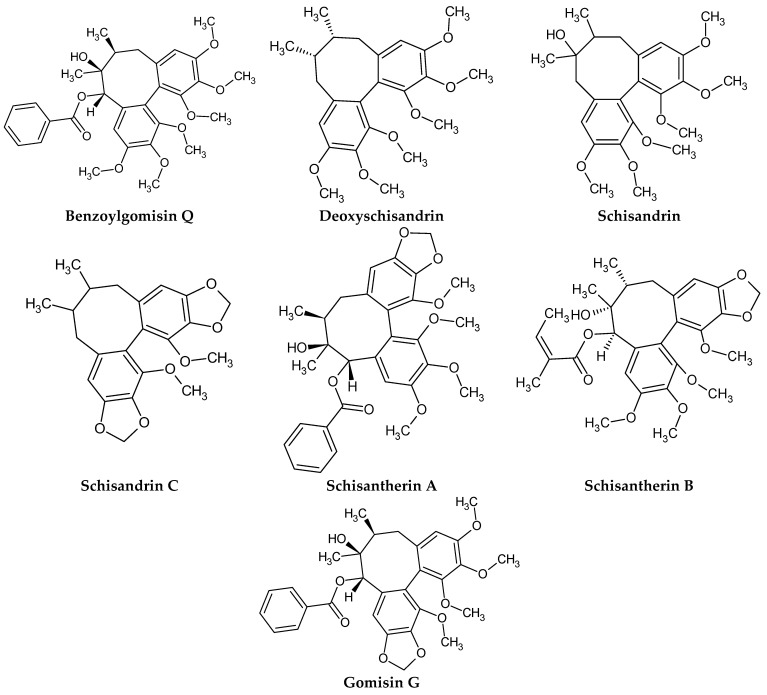
Chemical structures of dibenzocyclooctadiene lignans confirmed in *S. henryi*.

**Figure 4 molecules-28-04333-f004:**
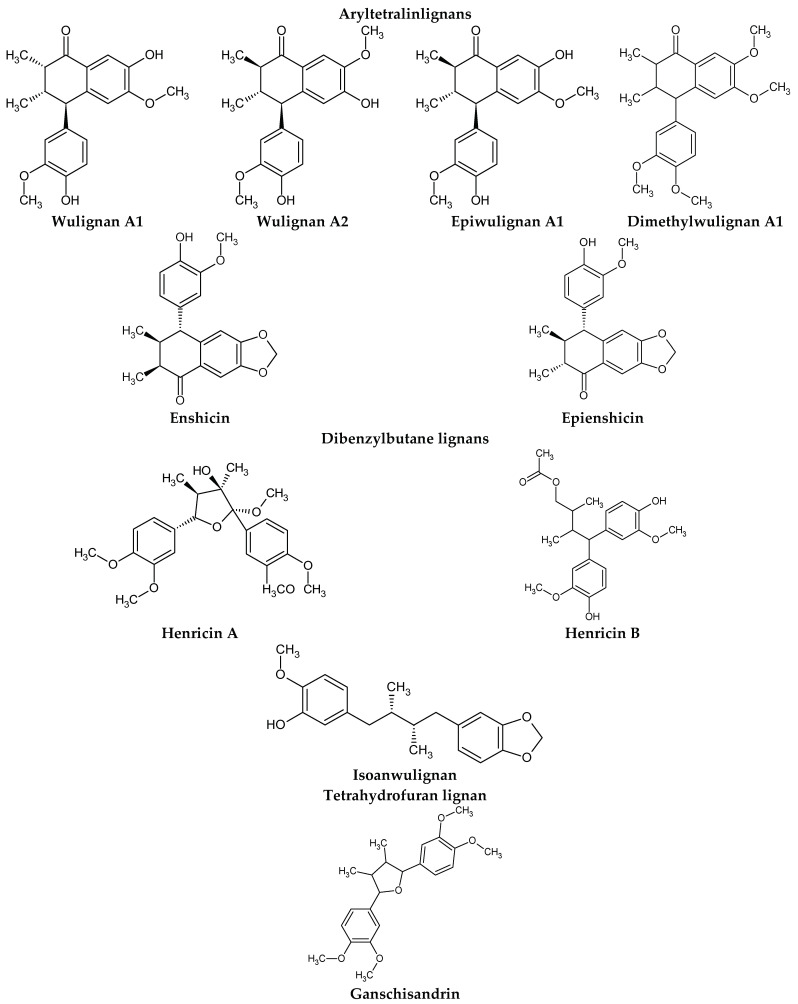
Chemical structures of aryltetralin, dibenzylbutane, and tetrahydrofuran lignans confirmed in *S. henryi*.

**Figure 5 molecules-28-04333-f005:**
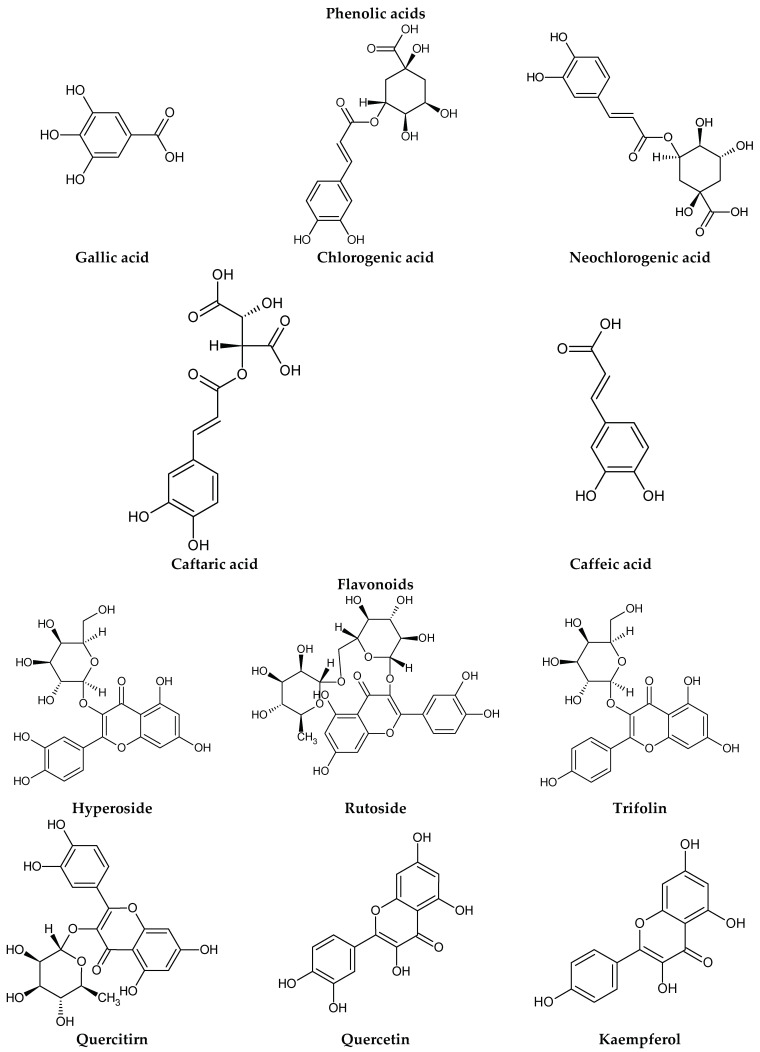
Chemical structures of other chosen compounds confirmed in *S. henryi*.

**Figure 6 molecules-28-04333-f006:**
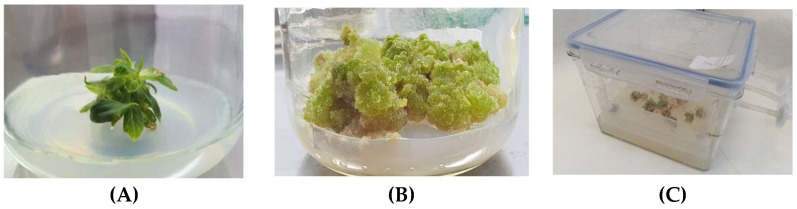
Types of selected in vitro cultures of *S. henryi*: (**A**)—agar microshoot cultures, (**B**)—agar callus cultures, (**C**)—microshoot cultures in PlantForm^TM^ bioreactor (photos were captured in the biotechnology laboratory of the Department of Pharmaceutical Botany, Collegium Medicum, Jagiellonian University, Kraków, Poland).

**Table 1 molecules-28-04333-t001:** Extraction and analytical methods applied for identification of compounds applied in previously published works on *S. henryi*.

Compound Extracted from Raw Material	Extraction Condition	Analysis Method	References
Enshicine from fruit	- the extract was dissolved in gasoline and then extracted with methanol - extracts were eluted with gasoline, benzene, benzene–ethyl acetate (10:1), benzene–ethyl acetate (4:1) and ethyl acetate	column chromatography on silica gel (gradient mode: benzene, benzene–ethyl acetate (10:1), benzene–ethyl acetate (4:1), and ethyl acetate)	[29]
Triterpenoids, lignans from leaves	- samples were extracted with methanol - extraction was carried out in an ultrasonic bath twice for 30 min, the extracts were centrifuged in a centrifuge	UHPLC-MS/MS with triple quadrupole mass filter (QQQ) (analytical column: Kinetex C18 150 × 4.6 mm, 2.6 µm, gradient mode: 50% methanol in water (A), 100% methanol (B) with 1% formic acid)	[32]
Phenolic acids and flavonoids from leaves	- samples were extracted with methanol - extraction was carried out in an ultrasonic bath twice for 30 min, the extracts were centrifuged in a centrifuge	HPLC-DAD (analytical column: Purospher RP-18, mobile phase: methanol and 0.5% acetic acid (A), methanol (B))	[32]
Triterpenoids from leaves and stems	- samples were extracted with 70% aqueous acetone (3 times, room temperature) - the extract was evaporated under reduced pressure - separation of the extract between water and ethyl acetate	column chromatography on silica gel, (chloroform–acetone (1:0 to 0:1), semi-preparative HPLC (analytical column: Agilent 1100 HPLC; Zorbax SB-C-18, Agilent, 9.4 mm 25 cm, gradient mode: methanol–water (65:35)	[48]
Nortriterpenoids form stems and leaves	- extraction with 80% aqueous acetone (3 times, room temperature) - filtrate was evaporated and the resulting residue was partitioned between water and ethyl acetate - the ethyl acetate layer was subjected to column chromatography on silica gel (chloroform/acetone gradient systems, 1:0 0:1 gradient systems)	RP-HPLC (55% methanol/water) column chromatography on silica gel (gradient systems: chloroform-Me2CO 1:0 0:1), repeated column chromatography (silica gel, petroleum ether/Me2CO, 9:1 and petroleum ether/ethyl acetate 4:1), RP-HPLC (gradient mode: 55% methanol/water)	[49]
Triterpenoids from stems	- extraction with 95% ethanol (4 times, room temperature) - evaporation ethanol under vacuum - the obtained filtrate was extracted with petroleum ether, ethyl acetate and n-butanol (4 times) - ethyl acetate extract applied to a silica gel column (petroleum elution)	column chromatography on silica gel (petroleum ether–ethyl acetate 4:1), repeated column chromatography on silica gel	[52]
Schinortriterpenoids from stems and leaves	- extraction with 70% aqueous acetone (3 times, room temperature) - the extract was distilled under reduced pressure under pressure to remove the acetone - the resulting filtrate was dissolved in water and separated with ethyl acetate - the ethyl acetate fraction was subjected to column chromatography with silica gel eluting with chloroform/acetone (1:0, 9:1, 7:3, 3:2, 1:1, and 0:1)	column chromatography with silica gel (chloroform/acetone 1:0, 9:1, 7:3, 3:2, 1:1 and 0:1), semi-preparative HPLC (analytical column; RP-18, Sephadex LH-20-methanol/water)	[50]

**Table 2 molecules-28-04333-t002:** Biological activity of selected dibenzocyclooctadiene lignans present in *S. henryi*.

Lignan	Chemical Structure of Compound	Maximal Content [mg/100 g DM ± SD]		Action	Mode of Action	Reference
		Microshoot Cultures	Leaves of the Parent Plant			
Schisandrin (schizandrin, schizandrol A, schisandrol A)	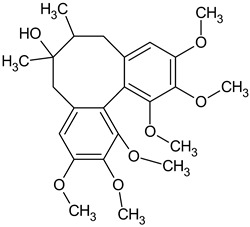	61.24 ± 0.23	8.62 ± 0.95	Antioxidant	- inhibits lipid peroxidation - causes a decrease in the expression of the MMP-1 protein	[54]
				Anti-osteoporotic	- increases the proliferation and activity of alkaline phosphatase in osteoblasts	[60]
Schisandrin C (wuweizisu C)	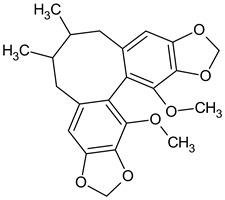	28.61 ± 0.23	1.06 ± 0.38	Antioxidant	- protects the skin from oxidative damage caused by UV-A radiation - increases the concentration of reduced glutathione - increases the level of α-tocopherol - increases the activity of antioxidant enzymes - increases the production of malondialdehyde	[57,62]
				Anti-inflammatory	- reduces the production of nitric oxide in RAW 264.7 cells - reduces mRNA expression and secretion of proinflammatory cytokines - blocks the production of protein kinase by inhibiting p38 mitogen	[57]
Deoxyschisandrin (schisandrin A, schizandrin A, deoxyschizandrin)	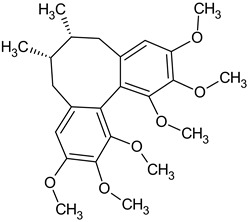	3.63 ± 0.27	1.70 ± 0.55	Anticancer	- inhibits the growth of OV-2008 and LoVo cell lines - induces apoptosis in cancer cells - inhibits the growth of cancer cells in the G2/M phase	[54,57,60,63]
				Antiviral	- inhibits the activity of DNA polymerase associated with HIV-1 reverse transcriptase - inhibits the replication of the HIV-1 virus	[63,64]
				Cardioprotective	- reduces arrhythmia - has a stabilizing effect on the activity of the heart - minimizes the risk of heart attack - reduces the release of MDA - reduces apoptosis - reduces the activity of caspase-3	
				Supportive treatment in intestinal dysfunction	- inhibits smooth muscle contraction - reduces BDNF protein expression in the colonic mucosa	[65]
				Anti-osteoporotic	- increases the proliferation and activity of alkaline phosphatase in osteoblasts	[61]
Schisantherin A (gomisin C, gomisin)	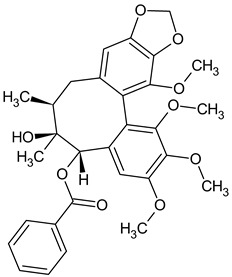	143.74 ± 0.43	4.75 ± 0.54	Anti-inflammatory	- reduces the concentration of iNOS and COX-2 in RAW 264.7 macrophages - inhibits the expression of the protein responsible for the phosphorylation of kinase, p38, and C-jun NH2-terminal kinase - inhibits the production of TNF-α - inhibits the production of NO and PGE2 - inhibits the expression of MMP-1, MMP-3, and MMP-13 - inhibits the inflammatory response in BV-2 microglia - inhibits NF-κB activation and p65-NF-κB gene translocation - reduces the level of IL-6 and IL-1β - inhibits the phosphorylation of nuclear transcription	[66,67,68,69]
				Anticancer	- inhibits the migration and proliferation of MKN45 and SGC-7901 cells - induces cell cycle arrest in the G2/M phase - induces JNK phosphorylation - has a cytotoxic effect	[59,70]
				Neuroprotective	- improves cognitive functions impaired by Aβ1-42 in mice - restores the activity of superoxide dismutase and glutathione peroxidase, Aβ1-42, malondialdehyde, and glutathione in the hippocampus and cerebral cortex - inhibits cytotoxicity in SH-SY5Y cells - increases the expression of Bcl-2 - activates PI3K/Akt survival signaling - protects against cytotoxicity induced by 6-hydroxydopamine - regulates the intracellular accumulation of ROS - inhibits the overproduction of NO - protects against MAPK, PI3K/Akt, and GSK3β activation	[60,71,72]
				Hepatoprotective	- relieves oxidative/nitrosative stress - reduces hepatocyte apoptosis - inhibits the protein kinase signaling pathway - alleviates pathological changes caused by thioacetamide - reduces serum levels of transaminases and hydroxyproline - reduces the expression of α-smooth muscle actin and collagen 1A1 proteins in the liver tissue - reduces the levels of TNF-α, IL-1β, and IL-6 - inhibits the proliferation and activation of HCS-T6 cells	[73,74]
				Cardioprotective	- reduces arrhythmia - regulates the activity of the heart - minimizes the risk of heart attack - reduces the release of MDA - reduces apoptosis - reduces caspase-3 activity	[64]
Schisantherin B (Gomisin B, Schisandrer, Wuweizi ester B)	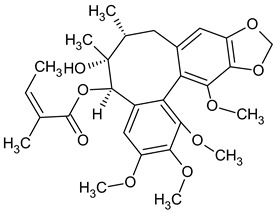	622.59 ± 0.57	48.99 ± 4.73	Antioxidant	- reduction in the level of superoxide radicals in neutrophils - influence on the inhibition of lipid peroxidation	[75,76]
Gomisin G	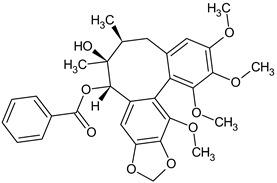	18.20 ± 0.18	1.62 ± 0.51	Anticancer	- reduces the viability of TNB C, MDA-MB-231, MDA-MB-468, and LoVo cell lines - inhibits AKT phosphorylation - reduces the amount of suppressor and phosphorylated Rb protein - decreases cyclin D1 level - induces apoptosis - affects the accumulation of cells in the sub-G1 phase - exhibits cytotoxicity to HeLa cells	[70,77,78]
				Antiviral	- inhibits HIV-1 replication	[79]

## Data Availability

Not applicable.

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
