# Peer review of "Schisandra henryi—A Rare Species with High Medicinal Potential"

_molecules, 2023, doi:10.3390/molecules28114333_

Round 1

Reviewer 1 Report

This review makes a comprehensive summary on Schisandra henryi as a species with high medicinal potential. This is clearly a hot subject of research worthy of a timely review. Generally, this review was well prepared with all important works on the reviewed field included. The useful information summarized in this work merits the publication in Molecules. The following minor concerns should be further addressed prior to acceptance.

1. The uniqueness of this review relative to the published reviews with a similar research topic on henryi and similar natural species remain unclear, which should be highlighted clearly in the abstract and introduction sections.

2. It will be great to include the extraction and purification methods for Schisandra henryi.

3. The authors should add more figures of notable examples in Section 5 to provide a clear presentation of the biological activities of Schisandra henryi for the readers.

4. Table 1 should include some quantitate data to support the qualitative statement.

5. In the last section of this manuscript (Future perspectives), please add more contents about the current problems that restrict application and clinical translation of Schisandra henryi for biomedical applications. In addition, please also point out future direction and solution to solve these problems. This will make this review more valuable and can provide the audience more information about what could be done in the future.

6. A TOC should be provided.

Author Response

REVIEWER 1

This review makes a comprehensive summary on Schisandra henryi as a species with high medicinal potential. This is clearly a hot subject of research worthy of a timely review. Generally, this review was well prepared with all important works on the reviewed field included. The useful information summarized in this work merits the publication in Molecules. The following minor concerns should be further addressed prior to acceptance.

  1. The uniqueness of this review relative to the published reviews with a similar research topic on henryi and similar natural species remain unclear, which should be highlighted clearly in the abstract and introduction sections.

Answer: Thank you for your comment. In the summary and introduction, information on the uniqueness of S. henryi compared to other species of the genus Schisandra was added. This comparison is mainly based on the presence of specific compounds in a given species. In S. henryi, apart from the characteristic Schisandra lignans, there are also lignans characteristic of this species, which are described in the text. Also compounds from the group of terpenoids, in particular schinortriterpenoids, which have recently been isolated from S. henryi, testify to the uniqueness of this species.

  1. It will be great to include the extraction and purification methods for Schisandra henryi.

Answer:  A new paragraph in chapter 3 and new table no 1 reviewed the performed so far studies on isolation and extraction were added.

  1. The authors should add more figures of notable examples in Section 5 to provide a clear presentation of the biological activities of Schisandra henryi for the readers.

 Answer: Chemical formulas were added to table 2

  1. Table 1 should include some quantitate data to support the qualitative statement.

Answer:  Quantitative results of lignans listed in Table 1 (now Table 2) were added

  1. In the last section of this manuscript (Future perspectives), please add more contents about the current problems that restrict application and clinical translation of Schisandra henryi for biomedical applications. In addition, please also point out future direction and solution to solve these problems. This will make this review more valuable and can provide the audience more information about what could be done in the future.

 Answer:  In the summary, information on the problems related to limiting the use of S. henryi extracts in medicine as well as problems related to biotechnological research in European countries were added. The focus was also paid also on clarifying the direction of further research on this species.

  1. A TOC should be provided.

Answer: TOC was added at the the beginning of manuscript.

Reviewer 2 Report

Please find my comments in the attached file

Author Response

Reviewer 2 comments molecules-2339446

Title: Schisandra henryi - a species with high medicinal potential

Two of the authors Halina Ekiert and Agnieszka Szopa have contributed previously to a review article concerning the same plant species and other related ones. https://doi.org/10.1007/s11101-018-9582-0

The recent review is more specifically focus on the plant species Schisandra henryi

According to dates of the cited references, only references 22 (by the authors), 40, 48, and 55 are dated after the previously published review, mentioned above.

Answer: The previous review focused on a comparative analysis of three species of the genus Schisandra, including Schisandra henryi. The study was aimed at comparing the chemical composition of the species, their place of occurrence and their use in modern science, including citing studies on biological activity.

The present article focuses only on the one least popular species, but with great medicinal potential - Schisandra henryi. The review presents the characteristics of this selected species. S. henryi is not a popular species, which is associated with a small amount of research on it. All available scientific literature data was included in this review. Structural formulas of compounds found in S. henryi are also presented, and biological studies on the group of compounds characteristic of the genus Schisandra - dibenzocyclooctadiene lignans - are described in detail. The article also presents all studies on the biological activity of S. henryi extracts from individual parts of the plant. Methods of extraction and isolation of compounds identified in S. henryi extracts were also developed. Efforts were made to show the information on the biotechnological studies performed by us on this species. Efforts were made to comprehensively approach the subject of S. henryi species.

- Accordingly, the title should be adjusted.

Answer: The title has been adapted to the content of the article.

- There should be a mention of the previous review in the abstract and or the introduction.

Answer: We added reference to a previous review that analyzed three Schisandra species, including S. henryi.

- What is the type of this review; narrative, comprehensive, …….. etc, please spedify.?

Answer: Information was added in the abstract that the review is comprehensive on the species S. henryi

Anyway, the following points should be fulfilled before accepting this article for publishing in Molecules.

Answer: The text has been proofread by a native speaker

Minor points:

The article needs careful English writing edition. Some, but not all, of the points need revisions are:

Line 17: studied → study.

Answer: The word in the text has been corrected.

Line 19: in vitro cultures use of to yield extracts. → of is redundant.

Answer: Thank you for the valuable tip, the phrase was removed from the text.

Figure 1. Please add the geographic location of the habitat of these pictures. Please use letters A and B to differentiate between left and right pictures.

Answer: Added the geographic location of the habitat where the plant in the photo grows

Figure 2. Chemical structures of selected dibenzocyclooctadiene lignans confirmed in S. henryi

why just structures of selected lignans → please comprehensively show all the structures for all the chemical classes isolated from the plant. This is a review. Isn’t it?

Answer: Thank you for your valuable attention, structural formulas of compounds detected in S. henryi have been added.

Line 380: Research in the field of plant biotechnology conducted on S. henryi → Although the plant in vitro culture is a technique of the biotechnology, the expression plant biotechnology is not consistent with the results discussed in this section.

Pleas change into “In vitro cultures of S. henryi”

Answer: The chapter title has been changed

Lines 381-383 is misleading, why the authors didn’t say shortly, in our plant tissue culture lab., the in vitro cultures of S. henryi were initiated for the first time (Figure 4).

Answer: Thank you. The text about biotechnology studies was improved.

Figure 4: Please add a reference.

Answer: We added information on where the photos were taken.

Line 407: Further studies on in vitro cultures of S. henryi are currently underway. Why the authors preferred to introduce a review although it was more suitable to provide their recent findings in an article then collect all research on the plant in a review.

Answer: The presented article only wanted to mention the possibilities associated with the use of one of the fields of plant biotechnology, which is the cultivation of in vitro cultures. The review was not intended to include all the results of the research team's in vitro S. henryi culture studies.

2

Major points:

The review lacks the main section: “Methodology”

In this section the authors should specify the used databases, the keywords, the covered period, the strategy of considering and excluding the papers.

Answer: Methodology section has been added as chapter 8.

Reviewer 3 Report

1.     The structures of all the compounds isolated from S. henryi and mentioned in the manuscript need to be shown in a figure.

2.     Biological activity of S. henryi extracts - review of scientific research: Most of the content in this section focuses on the activity of individual compounds, rather than just the activity of the extract.

3.   Biological activity of dibenozcyclooctadiene lignans present in S. henryi: Many compounds mentioned in this section are widely distributed in other plants of the genus Schisandra, and this section should focus on elucidating the activity of the characteristic compounds isolated from S. henryi.  

Author Response

REVIEWER 3

  1. The structures of all the compounds isolated from S. henryi and mentioned in the manuscript need to be shown in a figure.

Answer: Chemical structures have been added to the manuscript in the figure 5.

  1. Biological activity of S. henryi extracts - review of scientific research: Most of the content in this section focuses on the activity of individual compounds, rather than just the activity of the extract.

Answer: Currently, there is a small number of studies based on the biological activity of the whole S. henryi extract. All published studies on the biological activity of S. henryi extract are included in the text.

The work focused on lignans, which are one of the compounds isolated from the S. henryi extract due to their high therapeutic potential and great interest of researchers around the world.

  1. Biological activity of dibenozcyclooctadiene lignans present in S. henryi: Many compounds mentioned in this section are widely distributed in other plants of the genus Schisandra, and this section should focus on elucidating the activity of the characteristic compounds isolated from S. henryi.

Answer: Thank you for your comment. This chapter focuses on dibenzocyclooctadiene lignans, because this is the most numerous group of compounds found in extracts of both S.henryi and other species of the genus Schisandra, and thus there are a number of studies confirming their therapeutic effect, which was quoted in this article . Due to the low popularity of the species S. henryi, individual compounds characteristic of this species have not been tested for biological activity, which is why they could not be quoted in this review.

Reviewer 4 Report

The overview on Schisandra henryi presented in this work, specifically focused on the medicine potential, looks complete in the literature cited , but in in the present form is of poor interest for a general reader of Molecules, who expects to find aspects more profoundly related to chemistry, as indicated in the aims of the journal.

Therefore, to be accepted for publication in this jornal, the review should be completed in the following aspects, where applicable : discussion of metabolite biogenesis, relevant details on the isolation and structural elucidation of bioactive metabolites reported, especially as regards the absolute configuration assignment and stereochemical aspects, semi-synthetic and synthetic results.

In table 1 the molecular structures of the cited metabolites must be added.

Minor comments:

-          "review of scientific research" can be removed from the titles of paragraphs 4 and 5, because considered redundant in a review article

-          abbreviated title must be reported for all references.

Author Response

REVIEWER 4

The overview on Schisandra henryi presented in this work, specifically focused on the medicine potential, looks complete in the literature cited , but in the present form is of poor interest for a general reader of Molecules, who expects to find aspects more profoundly related to chemistry, as indicated in the aims of the journal.

Therefore, to be accepted for publication in this jornal, the review should be completed in the following aspects, where applicable : discussion of metabolite biogenesis, relevant details on the isolation and structural elucidation of bioactive metabolites reported, especially as regards the absolute configuration assignment and stereochemical aspects, semi-synthetic and synthetic results.

Answer: The figure (no.2) shows the scheme of lignan biosynthesis was added. Information on the isolation and derivatives of dibenzocyclooctadiene lignans has been added to the text.

In table 1 the molecular structures of the cited metabolites must be added.

Answer: Molecular structures were added.

Minor comments:

-          "review of scientific research" can be removed from the titles of paragraphs 4 and 5, because considered redundant in a review article

Answer: The given wording was removed as it was noted.

-          abbreviated title must be reported for all references.

Answer: In the References were adjusted to the Molecules style.

Round 2

Reviewer 2 Report

The authors have responded to the majority of the comments.

Some more points should be considered before accepting this work:

1- Figure 1. Please use letters A and B to differentiate between left and right photos.

2-Figure 2 quality is inferior.

please revise "Figure 2. The biosynthesis pathway of dibenzocyclooctadiene lignans" into "Figure 2. A suggested biosynthetic pathway of dibenzocyclooctadiene lignans [Reference]"

3- Drawing style and formats of the structures in Figures 3-5 should be identical. It needs intensive revision, for example, some structures are drawn based on their sterostructures, and some are not, the sugar part lack stereospecificity, and methoxy groups are drawn differently not only in the different molecules but also in the same molecule (Schisandrin). Font size should be identical as well (for example, font size of atoms in Schisandrin C and Schisantherin A structures)

4- Figure 5. Chemical structures of chosen other compounds confirmed in S. henryi. What was the rule to select only these structures, you probably want to respond to my comment (please comprehensively show all the structures for all the chemical classes isolated from the plant. This is a review. Isn’t it?), however, I don't favor this way of response. Please be more scientific.  

Author Response

Reviewer 2

Comments and Suggestions for Authors

The authors have responded to the majority of the comments.

Some more points should be considered before accepting this work:

  • - Figure 1. Please use letters A and B to differentiate between left and right photos.

Answer: Thank you for your valuable comment. Markings indicated by you have been added to the manuscript in Figure 1.

2-Figure 2 quality is inferior.

Answer: Thank you. Figure 2 has been re-added in enhanced quality.

please revise "Figure 2. The biosynthesis pathway of dibenzocyclooctadiene lignans" into "Figure 2. A suggested biosynthetic pathway of dibenzocyclooctadiene lignans [Reference]

Answer: According to your recommendation, the caption of Figure 2 was changed.

3- Drawing style and formats of the structures in Figures 3-5 should be identical. It needs intensive revision, for example, some structures are drawn based on their sterostructures, and some are not, the sugar part lack stereospecificity, and methoxy groups are drawn differently not only in the different molecules but also in the same molecule (Schisandrin). Font size should be identical as well (for example, font size of atoms in Schisandrin C and Schisantherin A structures)

Answer: Thank you very much for the valuable tip. The structures were corrected - all were formatted to the same size and the structures have been unified in terms of stereospecificity.

4- Figure 5. Chemical structures of chosen other compounds confirmed in S. henryi. What was the rule to select only these structures, you probably want to respond to my comment (please comprehensively show all the structures for all the chemical classes isolated from the plant. This is a review. Isn’t it?), however, I don't favor this way of response. Please be more scientific.

Answer: After the first review, the manuscript was supplemented with more formulas as noted by the reviewer. The choice of structures added was guided by the presentation of examples of compounds from each group of compounds confirmed in S. henryi. Moreover, we added suitable description of chemical studies on S. henryi in chapter 3 and in Table 2.

Reviewer 3 Report

1.     ‘Bicyclol’, is this compound a natural product? If so, please descript the source of the compound.

2.     Biological activity of S. henryi extracts - review of scientific research: Most of the content in this section focuses on the activity of individual compounds, rather than just the activity of the extract.

3.     The manuscript doesn’t reach the level of ‘Molecules’.

Author Response

Reviewer 3

Comments and Suggestions for Authors

  1. ‘Bicyclol’, is this compound a natural product? If so, please 3 the source of the compound.

Answer:

Bicyclol is a synthetic substance based on schisandrin C. In the initial stages of research on Schisandra chinensis extracts, chosen dibenzocyclocotadiene lignans were isolated and tested for hepatoprotective activity. Tests have shown that schisandrin C has promising activity in this manner. Despite many attempts, researchers have failed to elaborate full chemical synthesis procedure of schisandrin C, which forced them to continue research. It was found that appropriate changes in the positions of the methylene dioxide groups changing the length of the carboxylic acid to the biphenyl ring and changing the dicarnoxylate group to the hydroxyl group increase the effectiveness as well as bioavailability of the derivatives. In this way, bicyclol ((4,4′-dimethoxy-5,6,5′,6′-bis [methylenedioxy]-2-hydroxymethyl-2′-methoxycarbonyl biphenyl) was synthesized, which was registered as a drug by the Chinese FDA. The protocol of it synthesis as weel as the production method on the large scale have been covered by a patent [Hu, Wei.; Li,Y.; Zhang, C.Z. Enantioseparation of racemic antihepatitis new drug. Bicyclol with crystallization. Chin. Chem. Lett.,2005, 16(11), 1471-1473.; Liu, G.T.; Li, Y.; Wei, H.L.; Zhang, H.; Xu, J.Y.; Yu, L.H. Mechanism of protective action of bicyclol against CCl4-induced liver injury in mice. Liver Int., 2005, 25, 872-879; https://patents.google.com/patent/CN103242286A/en].

  1. Biological activity of S. henryi extracts - review of scientific research: Most of the content in this section focuses on the activity of individual compounds, rather than just the activity of the extract.

Answer: Referring to your comment, we would like to mention that there is a very small number of scientific studies on S. henryi extracts. All reports on the biological activity of S. henryi extracts are reported in this manuscript in the chapter 4. The work was enriched in the additional chapter focused on the bioactivity of chosen dibenzocyclooctadiene lignans, the presence of which was confirmed in S. henryi extracts (both qualitatively and quantitatively). We decided on this due to the fact that they are a group of compounds with the greatest therapeutic potential.

  1. The manuscript doesn’t reach the level of ‘Molecules’.

Answer: Thank you for your comment. Our work is the first summarizing data of chemical and biological studies on S. henryi. It’s rare but very interesting plant of high medicinal potential. We mentioned in our work about possibilities of in vitro cultivation of this plant species which is our object of interest too. The aim of our work was to show that S. henryi, like S. chinensis, could be a new raw material for medicinal use. We described in details the scientific studies on biological activities of S. henryi extracts. We decide to enrich the work with descriptions of studies on isolated lignans that are also present in S. henryi extracts. This may be a reason to undertake research in this direction in the future. We hope that our effort and innovative approach to research on new plant materials will gain acceptance. Additionally, we are confident that the work will be of interest to readers and will be cited by future researcher of species of the Schisandra genus.

Reviewer 4 Report

The revised version has been substantially improved. However, the authors have to make a further effort, especially as regards the absolute configuration assignment and stereochemical aspects. In details, in figure 5 the molecular structures of a series of metabolites are drawn without highlighting the configuration of the stereocenters. If known, it must be made explicit  as in the original publications and if there are some relevant aspects (i.e. particular methods of assigning the absolute configuration, determination via synthesis, etc.) they must be reported in the text.

Author Response

Reviewer 4

The revised version has been substantially improved. However, the authors have to make a further effort, especially as regards the absolute configuration assignment and stereochemical aspects. In details, in figure 5 the molecular structures of a series of metabolites are drawn without highlighting the configuration of the stereocenters. If known, it must be made explicit  as in the original publications and if there are some relevant aspects (i.e. particular methods of assigning the absolute configuration, determination via synthesis, etc.) they must be reported in the text.

Answer: Thank you for your comment. All chemical structures in figure 5 have been corrected for stereochemical aspects. Relationships have been transformed so that the configuration of the stereocenters is visible. The formulas were created on the basis of the PubChem database and using original publications, where chemical formulas of selected compounds were placed.
